# The Role of Protein SUMOylation in Human Hepatocellular Carcinoma: A Potential Target of New Drug Discovery and Development

**DOI:** 10.3390/cancers13225700

**Published:** 2021-11-14

**Authors:** Hongchao Yuan, Yuanjun Lu, Yau-Tuen Chan, Cheng Zhang, Ning Wang, Yibin Feng

**Affiliations:** School of Chinese Medicine, The University of Hong Kong, 10 Sassoon Road, Pokfulam, Hong Kong, China; hcyuan@hku.hk (H.Y.); yjlu@connect.hku.hk (Y.L.); ecyt@connect.hku.hk (Y.-T.C.); zttc@connect.hku.hk (C.Z.)

**Keywords:** hepatocellular carcinoma (HCC), post-translational modification, SUMOylation, cancer metastasis, cancer drug resistance

## Abstract

**Simple Summary:**

The small ubiquitin-like modifier is a highly conserved post-translational modification protein, mainly found in eukaryotes. Recently, studies have shown that SUMOylation promotes the development of liver cancer. This article summarises the recent literature on SUMOylation and Hepatocellular carcinoma (HCC). The mechanism of SUMOs in liver cancer cells was described. It also shows the potential of SUMO as a therapeutic target for liver cancer. At the same time, this article also enumerates the practical application in clinical, developing progress and future direction of HCC in clinical practice.

**Abstract:**

Small ubiquitin-like modifier (SUMO) is a highly conserved post-translational modification protein, mainly found in eukaryotes. They are widely expressed in different tissues, including the liver. As an essential post-translational modification, SUMOylation is involved in many necessary regulations in cells. It plays a vital role in DNA repair, transcription regulation, protein stability and cell cycle progression. Increasing shreds of evidence show that SUMOylation is closely related to Hepatocellular carcinoma (HCC). The high expression of SUMOs in the inflammatory hepatic tissue may lead to the carcinogenesis of HCC. At the same time, SUMOs will upregulate the proliferation and survival of HCC, migration, invasion and metastasis of HCC, tumour microenvironment as well as drug resistance. This study reviewed the role of SUMOylation in liver cancer. In addition, it also discussed natural compounds that modulate SUMO and target SUMO drugs in clinical trials. Considering the critical role of SUMO protein in the occurrence of HCC, the drug regulation of SUMOylation may become a potential target for treatment, prognostic monitoring and adjuvant chemotherapy of HCC.

## 1. Introduction

Hepatocellular carcinoma (HCC) is the most common primary liver cancer. As of 2020, HCC is the sixth most common cancer globally and the second leading cause of cancer-related deaths [1]. In the past few decades, the incidence of HCC-related deaths has increased. Globally, it is estimated that more than 90,000 new cases of HCC and more than 20,000 deaths related to HCC in 2020 [2]. The prognosis of hepatocellular carcinoma is poor, with a 3-year survival rate of around 16.6% [3]. Currently, only surgical resection and liver transplant resection are considered as potential cures for HCC. Sorafenib is the first tyrosine kinase inhibitor (TKI) approved by the FDA for the treatment of unresectable HCC. In the ten years before lenvatinib, sorafenib has been the main treatment for HCC [4]. Subsequently, Tyrosine Kinase Inhibitors (TKIs) drugs such as regorafenib, rammucirumab and cabozantinib have successively succeeded in clinical trials. The advent of these TKIs has brought a new approach to the second-line treatment of HCC. At the same time, immune checkpoint inhibitors (ICIs), combined immunotherapy and other influential treatment methods have also been introduced [5]. These drugs increase the options available to patients and improve the survival rate of HCC patients [6].

Similar to the ubiquitination pathway, SUMOylation is a post-translational modification process and plays a vital role in many cellular processes. Post-translational modification (PTM), referred to as the covalent and generally enzymatic modification of protein after protein biosynthesis, has been considered to play a definitive role in HCC progression and therapeutic treatments [7]. PTM is a vital cell signal transduction method. The ribosome translates mRNA into a polypeptide chain and forms a mature protein through PTM [8]. This process can introduce new functional groups, such as phosphate. Phosphorylation is the most common post-translational modification and a common mechanism for regulating enzyme activity [9]. Other post-translational modifications include glycosylation, lipidation… [10]. SUMOylation, as well as other PTMs, can change the properties of the target protein, including but not limited to their activity, positioning, stability and interaction with other proteins [11,12,13,14,15,16]. Therefore, the imbalance of PTMs can lead to various diseases [17,18,19,20].

The protein members of the SUMO family are covalently linked to the lysine residues of the specific target protein through an enzymatic cascade [21]. SUMOylation can either enhance the interaction of modified protein molecules by adding an interaction surface or mask the molecular interactions of the modified target protein. The target protein determines the specific consequences, which depends on the location, stability changes and activity of the target protein [22]. Abnormal SUMOylation modification of target protein is associated with many cancers including colorectal cancer, cervical adenocarcinoma and cervical adenosquamous carcinoma [23]. Blocking the SUMOylation of the target protein can inhibit the progress of cancer cells and stimulate interferon signal transduction, thereby further enhancing the tumour immune response of the subject [24]. A growing body of recent evidence has revealed that SUMOs-mediated SUMOylation was involved in multiple processes of HCC. Figure 1 [25,26] below is two examples that describe how SUMOylation affects the progress of HCC. Protein SUMOylation has been shown to regulate the cellular process related to cell proliferation and survival, migration and invasion and drug resistance of HCC. In this review, we summarised the recent research inputs of SUMO protein in HCC. Based on the current understanding, we discussed and envisioned the potential of clinical applications of HCC treatment acting on SUMOs. We hope that our article will help the identification of SUMOs targets.

## 2. The Dynamics of SUMOylation in HCC

SUMOs are a highly conserved protein family with a molecular weight of approximately 12 kDa that plays an essential role in the survival of eukaryote cells [27]. They share similarities with ubiquitin in structure and the C-terminal glycine (Gly) residue involved in the reaction [28]. Most of the SUMOs proteins contain a short consensus sequence ΨKXE, where Ψ represents a large hydrophobic amino acid, K stands for lysine, X represents any amino acid residue and E represents glutamic acid [21]. SUMOylation is highly dynamic and its cyclical process is shown in Figure 2 [29]. SUMOylation is one of the post-translational protein modification processes that is widely involved in regulating various aspects of protein and cellular activities. SUMOylation mainly regulates the interaction and positioning of the proteins [30]. Current research has found that more than one hundred proteins can serve as the substrate of the cellular SUMOylation process, most of which are nuclear proteins. A few foreign proteins can also be SUMOlated [31]. The main biological functions of SUMOylation modification include regulation of transcriptional activity, regulation mediated by promyelocytic leukaemia nuclear bodies(PML-NBs), the maintenance of the integrity of the genome by regulating the aggregation and separation of chromosomes, the process of DNA repair, regulation of protein stability and nuclear mass transportation, etc. [22].

There are five major protein subtypes in the SUMO family in mammals: SUMO1, SUMO2, SUMO3, SUMO4 and SUMO5. Family members are slightly different in length. SUMO1, SUMO2 and SUMO3 are expressed ubiquitously among species. SUMO1 is a polypeptide composed of 101 amino acids, which mainly modifies proteins in physiological states; SUMO2 and SUMO3 are composed of 101 and 95 amino acids, respectively. SUMO2 and SUMO3 have 97% sequence homology to each other, however, they are only about 50% homologous compared with SUMO1 [32]. They mainly modify proteins related to stress response [33]. The amino-terminus of SUMO2 and SUMO3 has a SUMO-binding conserved sequence, which can form SUMO multimers. SUMO multimers play a vital role in the intracellular localisation of target proteins and cooperative ubiquitination degradation. SUMO4, which is composed of 95 amino acid peptides, is only expressed in humans and pigs. It is mainly expressed in human kidneys and immune tissues (such as lymphatic tissues and spleen) [34]. SUMO5 is a recently discovered small ubiquitin-related modified protein. It is highly homologous to SUMO1 while characterised by two specificities: species specificity and tissue specificity. It is uniquely expressed in primates and actively transcribed in certain types of tissue, especially in testis and blood [35].

Similar to ubiquitination, SUMOylation is also a cyclical process, including growth, activation, binding and de-modification [36]. The SUMO gene is first expressed as an immature precursor protein [37]. This precursor protein has a short peptide with a length of approximately 2–11 amino acids at the C-terminus. Then the SUMO-specific enzyme Ubl-specific proteases (Ulps) cuts off the short peptide at the C-terminal end, exposing the Gly residue [38] and converting it into a mature functional SUMO protein. In mature SUMO protein-mediated SUMOylation, an isopeptide bond is formed between the C-terminal Gly residue and the Lys ε-amino group of the target protein.

The molecular cascades of protein SUMOylation involve multiple enzymes that similarly function in protein ubiquitination: E1 activating enzyme, E2 binding enzyme and E3 ligase, but the enzymes involved in the two reaction pathways are entirely different [38]. Bosis and Melchior [39] showed that a low concentration of reactive oxygen species (ROS) might recruit the SUMO E1 subunit Uba2 and E2 enzyme Ubc9 to catalyse the formation of disulphide bonds by cysteine, thereby preventing the formation of Ubc9-SUMO thioesters and inhibiting SUMO engages. SUMOylation is a dynamic and reversible modification process. The SUMO modification system can be regulated by regulating the expression of each component of the SUMO modification pathway and the activity of the SUMO enzymes. Extracellular stimuli can signal to the nucleus, triggering the rapid recruitment of the SUMO E3 ligase promoter, leading to immediate transcription repression. In addition, there is a specific protease in the cells that can remove SUMO from the modified protein, including de-SUMO enzymes in mammals, called SENP. They have different positioning, slightly different functions and different substrate specificities. SENP1 and SENP2 can separate SUMO1 and SUMO2/3 proteins. SUMO2/3 proteins mainly separate SENP3 and SENP5. SENP6 and SENP7 mainly decompose the multi-chain of SUMO2/3 [40]. It has been reported that SENPs are critical regulators of the SUMOylation regulatory mechanism. A study by Cui and Wong [41] showed that SENP1 significantly increased the stability and transcriptional activity of hypoxia-inducible factor 1 α (HIF-1α) under hypoxic conditions through deSUMOylation [42].

Other post-translational modification events, such as phosphorylation, ubiquitination and acetylation, can regulate SUMOylation through cross-talk [43]. Phosphorylation can regulate the binding of SUMO to the target protein through the motif PDSM (phosphorylation-dependent SUMO motif) [44]. This phosphorylation-dependent SUMOylation regulation is called phosphorylation-SUMOylation switch. Similarly, in proteins such as the transcription factor Elk-1, an extended SUMO modification motif has been found, containing a cluster of acidic residues downstream of the SUMO core modification site [45]. The dependence of SUMO and substrate Conjugation of amino acid residues. Studies have shown that SUMO coupling can occur with the ubiquitin in specific proteins on the same lysine residue. For example, the competition between SUMOylation and ubiquitination on the same lysine residue regulates the stability of IκBα [46].

It should be noted that SUMOylation and ubiquitination are not entirely competitive. In some cases, SUMOylation can provide a signal to ubiquitin ligase [47]. For example, in the process of post-translational modification of the MEF2 protein, the SUMOylation-acetylation switch is regulated by phosphorylation. These studies show the importance of signalling cross-talk in regulating protein SUMOylation [47].

## 3. The Expression and Clinical Significance of SUMOs Proteins in HCC

At present, there have been studies expounding the possible clinical applications of SUMOs protein. In the study of Jiann et al. [48], it was found that the mRNA levels of SAE1 in liver cancer tumour tissues were significantly upregulated (*p* < 0.0001). Patients with higher SAE1 mRNA have a worse disease survival rate (DSS) and progression-free survival rate (PFS). This is consistent with the role of SUMOylation proteins such as SUMO1 and Ubc9 in HCC. Research further shows that SAE1 is closely related to hypoxia and impaired metabolism [49]. Highly expressed SAE1 upregulates reactive oxygen species (ROS), glycolysis and cholesterol homeostasis pathway, indicating the potential of SAE1 as a biomarker for the prognosis of HCC [50]. In addition, Jennifer et al. discovered and produced a catalytically inactive recombinant fragment of the U.D. domain of KmUlp1, named KmUTAG [51]. KmUTAG can effectively and extensively bind to proteins in the SUMO family. The combined compound is highly stable and will not denature the protein even under extreme conditions. Through these experiments, they revealed the essential details of binding to the target protein in SUMOylation. More importantly, they predict that the recombinant fluorescent KmUTAG can visualise the distribution and flow of SUMO and SUMOylated proteins in living cells. It provides new ideas and methods for future research on the role of SUMOylation in HCC.

We extracted the mRNA expression levels of SUMO1, 2 and 3 in HCC tumour tissue and normal liver tissue, respectively from the GEPIA database. As shown in Figure 3 [52]: The expression of different SUMOs in HCC was retrieved from the GEPIA (http://gepia.cancer-pku.cn/, accessed on 3 November 2021) database. The expression levels of SUMO1, SUMO2 and SUMO3 in liver cancer cells were significantly higher than those in normal cells (*p* < 0.01). This indicates that the SUMOs protein plays an essential role in the development of HCC and suggests the potential of SUMOs as future targets. These data and Jiann’s research results are mutually corroborating, implying the potential of SUMOs as prognostic markers of HCC [48].

As shown in Figure 4 [52], we have also illustrated the relationship between the expression of SUMOs mRNA and the prognosis of HCC patients from GEPIA. The GEPIA (http://gepia.cancer-pku.cn/, accessed on 3 November 2021) database shows that the expression level of SUMOs has a significant impact on the overall survival(OS) time of patients. As the figures are shown below, whether it is overall survival or disease-free survival (DFS). Patients with low expression levels of SUMOs have better prognosis and survival rates. These results suggest that SUMOs are abnormally expressed in human tumour tissues and SUMOs may be used as a new malignant tumour marker.

## 4. The Role of Protein SUMOylation in HCC Progression

### 4.1. SUMOylation in HCC Carcinogenesis

Protein SUMOylation has been shown to maintain the steady-state balance of protein functions in normal tissues and various tumours. Increasing evidence indicated that the SUMO enzyme participates in carcinogenesis through a series of complex HCC mainly originates in the inflammatory environment of the (See Table 1 below).

HCC mainly originates in the inflammatory environment of the liver. Extensive studies have shown that SUMO protein is an essential pathway for activating NF-κB [50,53,54,55,56]. SUMO protein can co-localise with P65 and affect the interaction of P65 with other proteins [56,58]. In short, SUMO protein can promote the canceration of liver cells in an inflammatory environment. Yen et al. found that TNF-α-mediated SUMO-1 modification of CPAP is significant in the activation of NF-κB [67]. HBx can upregulate CPAP at the transcriptional level by interacting with CREB [56]. CPAP can interact with HBx to provide a microenvironment for the development of cancer cells. Inhibition of CPAP can inhibit the proliferation and migration of HCC. SUMOylated HBx enhances the interaction with CPAP and thereby promoting the carcinogenic effect of HBx.

In addition to early carcinogenic events, it is evidenced that SUMO protein is essential in maintaining the carcinogenicity of HCC. Nuclear factor erythroid-2 related factor 2 (NRF2) is a transcription factor that can promote the progression of various cancer [77]. The NRF2 pathway can maintain redox homeostasis in the cell, making it indispensable in normal cells [78]. Compared with normal cells, cancer cells have unique metabolic characteristics. For example, to accelerate the synthesis of biological substances to achieve rapid value-added or improve tolerance to oxidative stress. The research of Guo and Xu showed that NRF2 is a crucial transcription factor that maintains cell redox homeostasis and promotes the phenotype of malignant tumours [78]. Reports have shown that NRF2 plays a key role in the metabolism of cancer cells. Abnormal activation of the NRF2 pathway in cancer cells leads to reprogramming of intermediate metabolism, supporting cancer cell proliferation and tumorigenesis. The SUMO protein regulates the functionality of NRF2. Inhibiting the SUMOylation of NRF2 can slow down the carcinogenesis of hepatocellular carcinoma. SUMO protein-mediated NRF2 SUMOylation can enhance the clearance rate of intracellular nutrient activity and promote the synthesis of serine-free HCC. In addition, HCC lacking serine increases the SUMOylation of NRF2 and promotes the process of HCC [50]. In short, SUMOylation of NRF2 can maintain the carcinogenicity of HCC in liver cancer cells and promote the growth and proliferation of HCC cells. At the same time, it can improve the tolerance of HCC cells to oxidative stress.

### 4.2. SUMOylation in the Proliferation and Survival of HCC Cells

The SUMOs signalling cascade is critical to gene expression, genome integrity and cell cycle progression [79]. The recent development of small molecule inhibitors has made possible the therapeutic targeting of the SUMO pathway. Blocking SUMOylation not only reduces the proliferation of cancer cells but also enhances the anti-tumour immune response by stimulating interferon signals, which indicates that the SUMOylation inhibitor has a dual mode of action (Table 1).

p65 is one of the most critical subunits of NF-κB and a key regulator in the development of HCC [80]. Studies have shown that p65 can be modified by exogenous SUMO3 and expressed in Human Embryonic Kidney Cells 293 (HEK-293) cells [58]. Liu and colleagues found that in HCC adjacent tissues, SUMO2/3 is expressed in large amounts in the cytoplasm [61]. However, in contrast, the level of SUMO2/3 in tumour tissues was down-regulated. Consistent with this result, the expression of p65 is also upregulated in adjacent tissues. Consistent with SUMO2/3, it is mainly located in the cytoplasm. This means that SUMO2/3 is closely related to the expression of p65 in liver tissue and is co-localised. The experiment further confirmed this conclusion by interacting with p65 and SUMO2/3 through immunoprecipitation and double-labelled immunofluorescence (r = 0.800, *p* = 0.006) [61]. In addition, Liu et al. found that SUMO2/3 had a dose-dependent up-regulation of cytoplasmic p65 protein levels but did not affect its mRNA levels. At the same time, TNF-α induced an increase in the binding of SUMO2/3 to p65, accompanied by a decrease in ubiquitin binding to p65. Further studies have shown that the overexpression of SUMO2/3 reduces the proliferation ability of liver cancer cells but does not affect the migration of liver cancer cells.

The STAT4 (PIAS4) protein regulates various biological activities, including post-translational modifications of the protein, such as SUMOylation. PIAS4, as an E3-SUMO ligase, can inhibit the functions of many proteins such as AMPKα and NEMO [81]. In the study of Liu and Zhou et al., the role of PIAS4 in hepatocellular carcinoma was explored [24]. To analyse the expression of PIAS4 in 38 cases of HCC patients with cancer tissues and adjacent tissues and its relationship with the prognosis of patients. The results showed that patients with upregulated PIAS4 levels in liver cancer tissues had a poor prognosis (*p* < 0.05). Highly expressed PIAS4 promotes the SUMOylation of AMPKα and NEMO. This leads to the proliferation, migration and invasion of HCC cells. This indicates that PIAS4 promotes the progress of HCC by promoting SUMOylation of and NEMO [24].

Large tumour suppressor genes (Lats) can enhance the homeostasis of cells. Furthermore, it can mediate Hippo’s inhibitory signal [82]. Liu and Mei et al. found that SUMO of Lats1 affects its kinase activity, especially Hippo signal transduction [64]. SUMO1 interacts directly with Lats1 and the loss of SUMO pathway function interferes with the SUMOylation of Lats1. The SUMOylation of Lats1 reduces the phosphorylation of Lats1 through antagonism, leading to the weakening of Lats1 kinase activity and inhibition of Hippo signalling. In addition, more SUMO-Lats1 conjugates can be detected in cancer cells. Eventually, promote the proliferation and survival of liver cancer cells.

### 4.3. SUMOylation in the Migration, Invasion and Metastasis of HCC

SUMO proteins are also necessary for the metastasis and invasion of HCC. It is mainly achieved through the signal pathway that affects NF-κB [24,25,66,72]. Interestingly, SUMO protein can promote and inhibit the metastasis of HCC. This reflects the diversity of SUMO protein substrates and the complexity of SUMO protein in the HCC process.

During endoplasmic reticulum stress, the secretion of astrocyte-derived neurotrophic factor (MANF) from the right to midbrain can be upregulated. MANF can interact with P65 to inhibit P65-mediated inflammation [25]. Liu and Wu’s research found that: in HCC tissues, the mRNA and protein levels of MANF are significantly lower than those of adjacent tissues [25]. Follow-up of the prognostic level of patients found that the inflammatory microenvironment in the tumours of patients with low MANF levels is more extensive. Patients with higher levels of MANF have better disease-free survival and overall survival. At the same time, in vitro experiments show that MANF can also inhibit the metastasis and invasion of liver cancer cells. MANF promotes the hepatocyte-specific depletion of HCC induced by n-nitrosodiethylamine (DEN) by upregulating the level of Snail1/2 and promoting the epithelial-mesenchymal transition (EMT) of HCC cells. MANF is located in the nucleus and co-localises with liver cancer cells treated with p65 and tumour necrosis factor-α (TNF-α) in liver cancer tissues. SUMO1 mediated MANF SUMOylation can promote the nuclear translocation of MANF and enhance the interaction between MANF and p65, thereby inhibiting the growth and metastasis of HCC.

In Liu and Tao’s experiments, the interaction of SUMO1 and P65 accelerated the process of HCC. P65 is upregulated in liver tumour tissues and is related to liver cancer [30]. SUMOylation and regulation of various intracellular processes, such as nuclear import of targeted proteins. The nuclear transport of p65 leads to the activation of NF-κB and p65 contains multiple SUMO interacting sequences (SIM). The potential role of SUMO1 in HCC is demonstrated by regulating the subcellular localisation of p65. The results showed that in the tumour tissues of HCC patients, the positive immune response of SUMP1 and P65 was significantly higher than that of adjacent tissues. The data showed that there was a positive correlation between the positive immune response of SUMO1 and P65 (r = 0.851, *p* = 0.002) [30]. It is speculated that the up-regulation of TNF-α and the hypoxic environment both upregulate the SUMO1-modified P65 SUMOylation. SUMO1 can upregulate the nuclear translocation of P65 and promote the transcriptional activity of NF-κB.

The above two studies have shown that SUMO1 can co-localise with P65, thereby affecting cell function. Nevertheless, it is interacting with different objects and led to the opposite result. This shows that the SUMOs protein is not limited to a simple factor determining the fate of HCC. The mechanism and critical points need to be further studied.

### 4.4. SUMOylation in the Tumour Microenvironment of HCC

The tumour and the supportive tumour microenvironment (TME) interact and promote each other [83]. The structure of TME includes tumour-infiltrating monocytes/macrophages, immunosuppressive cells, fibroblasts, blood vessels and secreted inflammatory factors [84]. In TME, liver cancer cells can target and perceive stromal cells and hijack the surrounding normal cells to support their development [85]. Tumour-infiltrating macrophages are one of the most abundant stromal cell types in HCC TME. They inhibit anti-tumour immunity and secrete various inflammatory mediators by inducing extracellular matrix remodelling, angiogenesis, metastasis and treatment resistance. Promote tumour progression [85]. These macrophages are mainly derived from circulating monocyte precursor cells and differentiate into mature macrophages under regulatory factors derived from tumour cells. Tumour-derived extracellular vesicles are essential mediators of cell-to-cell communication during tumorigenesis and development [86]. Hou et al. proved that extracellular bodies derived from hepatocellular carcinoma rebuild the tumour microenvironment and promote the progression of liver cancer in a PKM2-dependent manner [73]. HCC-derived exosomes PKM2 induce the metabolic reprogramming of monocytes and the phosphorylation of STAT3 in the nucleus, thereby upregulating differentiation-related transcription factors, leading to monocyte-macrophage differentiation and tumour microenvironment remodelling. In liver cancer cells, SUMOylation of PKM2 induces plasma membrane targeting and exosomal excretion through interaction with ARRDC1. Cytokines/chemokines secreted by macrophages strengthen the connection of PKM2-ARRDC1 in liver cancer in a CCL1-CCR8 axis-dependent manner, further promoting the secretion of PKM2 in the liver cancer cells, and forming a feedforward regulatory loop. Clinically, extracellular PKM2 can be detected in the plasma of patients with hepatocellular carcinoma. This study highlights the mechanism by which extracellular PKM2 reshapes the tumour microenvironment and reveals the possibility of extracellular PKM2 as a diagnostic marker for hepatocellular carcinoma.

### 4.5. SUMOylation in HCC Drug Resistance

The low cure rate and poor prognosis of HCC are closely related to the easy development of drug resistance in HCC [3]. There are many reasons for the development of drug resistance in HCC. Such as the specific hypoxic microenvironment in solid tumours, mutations of tumour suppressor genes, or abnormal expression of cell membrane transporters [3]. These factors will lead to the evolution of tumour cell icons, structure and function. This leads to HCC insensitivity to drugs, drug resistance and chemotherapy failure [87]. As mentioned above, hypoxia is an essential feature of HCC. Under hypoxic conditions, the SUMOylation of GLI proteins can be upregulated [74]. GLI is an essential mediator of the SHh pathway signal transfer [88]. SHh can promote the drug resistance of liver cancer cells, so it has become an essential target for the development of drug resistance. Zhang et al. found that under hypoxic conditions, Ssd can inhibit the malignant characteristics of liver cancer cells. In addition, it can also enhance the sensitivity of HSV, GCV and other chemotherapeutic drugs [74]. At the same time, Ssd can promote SENP5-mediated GLI de-SUMOylation and down-regulate the expression and interaction of SUMO1 and GLI. Thereby slowing down the drug resistance of HCC and enhancing the effect of chemotherapy. The adjustment of Ssd depends on time and dose. The higher the dose, the longer the action time and the stronger the inhibitory effect on GLI and SUMO1.

UBC9 is the only consistent SUMO protein E2 enzyme [89]. It is an essential protein in the SUMO-mediated Summarisation process. It plays a vital role in the cell cycle, apoptosis regulation, DNA damage repair and gene transcription [90]. At the same time, it is also crucial in the nuclear and cytoplasmic transport of proteins [91]. Reports show that overexpression of UBC9 can promote drug resistance in breast cancer cells [92,93]. In the experiment of Fang and Qiu, the mechanism of UBC9-mediated SUMOylation on chemotherapy resistance of liver cancer cells was explored. Experiments have found that the expression of UBC9 in HCC is significantly upregulated compared with adjacent tissues [60]. The magnitude of up-regulation is positively correlated with tumour size and density. After down-regulating the expression of UBC9 through shRNG overexpression, the drug resistance of cancer cells was inhibited. It indicates that UBC9 may be related to the drug resistance of liver cancer cells. Silencing the SUMO E2 gene UBC9 may inhibit the generation of HCC resistance and is a potential target for the treatment of HCC.

### 4.6. SUMOs Related Drugs

Growing evidence has shown that SUMO protein plays a crucial role in the entire process of HCC. Potentially, the regulation on the expression of SUMO protein may interfere with the process of HCC, which makes SUMO a promising target for HCC treatment. At the same time, different expression levels of SUMO protein can regulate the sensitivity of HCC to chemotherapy and have a certain impact on HCC resistance [94]. The expression of SUMO protein is also related to the prognosis and can be used as an indicator to detect the prognostic effect. However, there is no current SUMOylation-related drug on the market. Only one SUMO-related drug is undergoing clinical trials. SAE (SAE1, SAE2, UBA2) is the activating enzyme of SUMO. In the enzyme cascade of SUMOylation, the ATP-dependent process catalysed by SAE can activate the SUMO protein [23]. TAK-981 is an SAE-based SUMOylation inhibitor [95]. A SUMO-TAK-981adduct can be formed as a site to inhibit SUMOylation (NCT number: NCT03648372, NCT04065555, NCT04074330, NCT04776018 and NCT04381650). The clinical trials of TAK-981 are still in progress, so follow-up data is needed for discussion. In addition, the effect of TAK-981 in HCC is currently poorly understood and further testing is needed.

## 5. Discussion

As mentioned above, there are currently no commercially available SUMO drugs, studies have shown that some natural compounds can regulate the SUMO process. For example, according to the study of Fukuda et al., ginkgolic acid and its analogues have an inhibitory effect on SUMOylation and do not affect ubiquitination in the body [96]. Ginkgo acid was directly bound to SAE1 and can inhibit the formation of E1-SUMO intermediates [97] and thus inhibited SUMOylation in the body. These studies will provide valuable tools for studying the role of SUMO conjugates in various pathways in cells and provide a basis for the development of drugs for diseases involved in abnormal Summarisation. Similarly, another natural compound named Kerriamycin B was reported to act as a SUMOylation cascade inhibitor by explicitly binding to E1 [98].

Spectinomycin B is a natural antibiotic. Unlike ginkgolic acid or kerriamycin B, spectinomycin B1 directly binds to E2 (Ubc9) and selectively prevents the formation of the E2-SUMO intermediate. In other words, Ubc9 is the direct target of spectinomycin B1 [99].

Triptolide is another natural modulator of the SUMOylation cascade [100]. According to the research of Huang and colleagues, Triptolide can down-regulate the mRNA and protein expression levels of SENP1, thus inhibiting the occurrence of deSUMOylation and maintaining the level of SUMOylation. Its anti-tumour activity may be related to the downregulation of SENP1 to restore the balance of SUMOylation and de-SUMOyaltion. Momordin Ic (Mc) is another SUMOylation cascade modulator for SENP1. Mc reduces the cleavage of SUMO2-ΔRanGAP1 and also changes the thermal stability of SENP1. Consistent with this, Mc increased the level of SUMOylated protein [101]. Indeed, Mc has potential therapeutic value for prostate cancer, but its role in HCC needs further research.

Unfortunately, there are currently no detailed data on the specific effects of these natural compounds in HCC. Most of the experiments are done in vitro or with the virus. Only momordin-Ic has been shown to inhibit the growth of prostate cancer cells in vivo and in vitro [101]. But the performance of momordin-Ic in HCC is not yet researched. In addition, the inhibitory effects of these natural compounds on SUMOs have not yet been quantified. Only ginkgolic acid has been shown to have a strong inhibitory effect on SUMOylation. In 72 h of treatment, 10μM of ginkgolic acid can significantly inhibit the activity of SUMO-1 [102]. The natural compounds found so far that can interact with SUMOs proteins are all SUMOylation inhibitors. The use of these compounds can better study the significance of inhibiting SUMOylation in HCC. At the same time, according to current research, these SUMOs inhibitors also can become an adjuvant therapy for HCC.

## 6. Conclusions

SUMO modification is an essential post-translational modification, which has been shown to play a crucial role in various processes of HCC. This article reviews the role of SUMOylation in HCC carcinogenesis, proliferation and survival, metastasis and invasion, tumour microenvironment and HCC drug. In general, SUMO1 and SUMO2/3 have a promoting effect in all stages of HCC development. SUMOylation rarely directly promotes the development of cancer. It is an indirect way to enhance the carcinogenic effect of its target proteins. For example, co-localisation with carcinogenic factors (LKB1), strengthening the stability of carcinogenic factors (β-catenin), or a necessary condition for carcinogenic factor activity (Cbx4). It is worth noticing that SUMOs do not always promote the development of HCC. For example, SUMO1-mediated MANF SUMOylation can enhance the interaction between MANF and p65, thereby inhibiting the growth and metastasis of HCC. Overall, recent studies have shown the potential of SUMO1 and SUMO2/3 as therapeutic targets and prognostic markers for HCC. A drug-related to SUMO, TAK-981, is undergoing clinical trials. Nevertheless, there is currently no data for TAK-981 and HCC. Moreover, studies have shown that some natural compounds can regulate the expression of SUMO, but this regulation has not been confirmed in the liver. Translating these potentials into treatments and verifying the effects of these potentials in clinical treatment may be the focus and direction of future research.

## Figures and Tables

**Figure 1 cancers-13-05700-f001:**
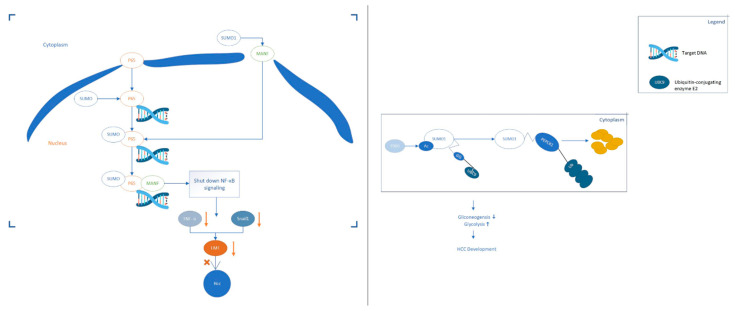
Two examples of SUMO proteins affecting the development process of HCC. (**Left**): SUMO1 binds to MANF and promotes the transfer of MANF to the nucleus. The SUMOylation of P65 can up-regulate the combination of P65 and MANF to form a repressor complex, thereby hindering the transmission of NF-κB signals. Eventually, the expression of TNF-α and Snails downstream of the NF-κB signalling pathway is down-regulated, which hinders the progress of HCC. (**Right**): The expression of acetylase p300 in liver cancer is up-regulated and p300-mediated acetylation of Ubc9 can enhance its binding to PEPCK1. Increase the SUMOylation level of PEPCK1. Promote the degradation of PEPCK1, thereby accelerating the proliferation of liver cancer cells.

**Figure 2 cancers-13-05700-f002:**
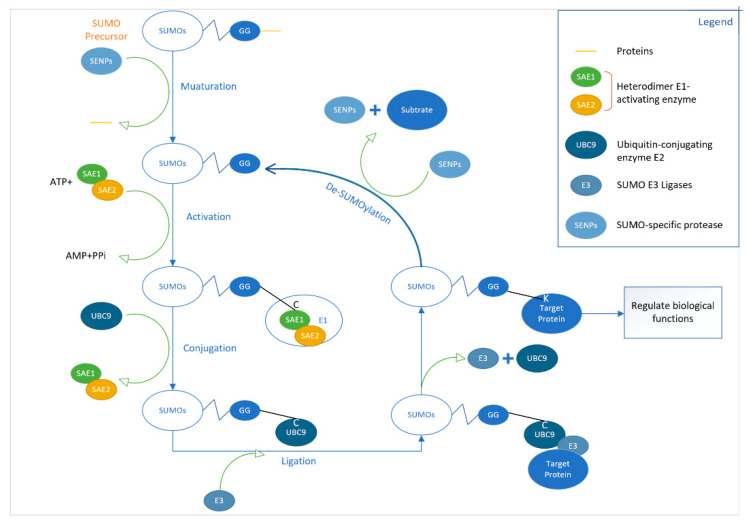
Schematic diagram showing the pathway related to protein SUMOylation. Maturation/de-SUMOylation: SENPs cleave the SUMOs protein from the precursor protein to produce mature SUMO protein; Activation: A thioester bond is formed between the GG of the SUMO protein and the cysteine residue of SAE1/2; Conjugation: A thioester bond is formed between the GG of SUMO protein and the cysteine residue of Ubc9; Ligation: Ubc9 and E3 ligases catalyse GG and target substrate lysine residues to form isopeptide bonds.

**Figure 3 cancers-13-05700-f003:**
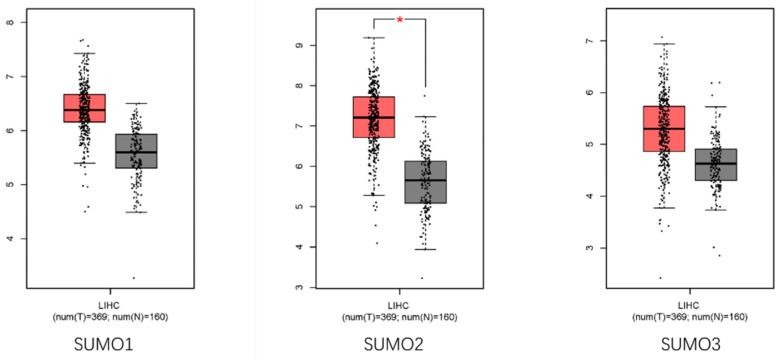
The expression differences of SUMOs mRNA in human HCC specimens. (* *p* < 0.01, The ordinate is the expression amount represented by log2(TPM + 1)).

**Figure 4 cancers-13-05700-f004:**
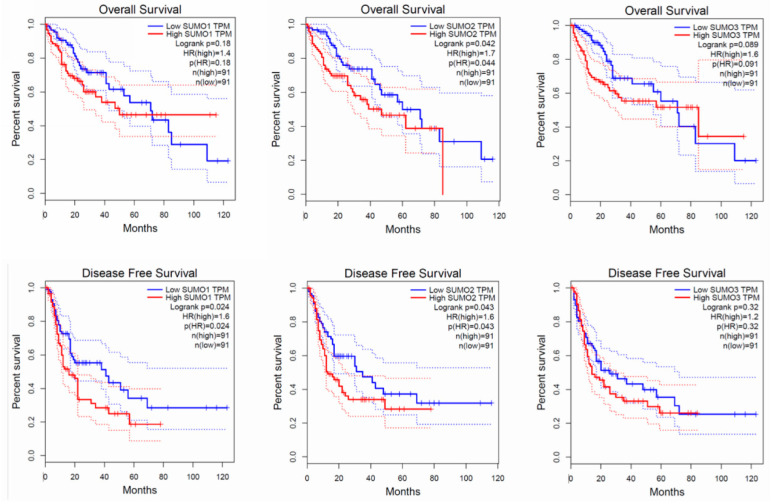
The prognosis value between the expression of SUMOs mRNA. The red line in the figure below represents patients with high expression of SUMOs mRNA and the blue line represents patients with low expression. Use 75% as the high cut-off and 25% as the low cut-off. Patients with low expression of SUMO protein had better OS and DFS than patients with high expression.

**Table 1 cancers-13-05700-t001:** The Role of SUMOs in HCC progression.

Actions	Models	Regulation/ Pathway	Effects	Reference
HCC carcinogenesis	In Vitro, MHCC97H, MHCC97L, MHCC-LM and QGY7703Hep3B, HepG2, Huh7, SMMC-7721, BEL-7402, BEL-7404. In vivo, male BALB/c nude mice (5 weeks old)	SIRT1-FTO	SUMO ligase E3 RANBP2 promotes SIRT1-related FTP down-regulation	[53] 2020
	in vitro, HepG2	IQGAP2	SUMOylation inhibits AKT phosphorylation through cross-talk, thereby inhibiting the expression of HBV genes.	[54] 2019
	In vitro. HEK293T, SMMC-7721 and HepG2 cells	ROS&PHGDH	SUMOylised NRF2 can upregulate the synthesis of serine in HCC.	[50] 2019
	In vitro, SMMC-7721, HepG2 and SMMC-7721-shShp2	ERK	Shp2 SUMOylation upregulates the activation of ERK	[55] 2015
	In vitro, HepG2, HepG2X, Hep3B, SK-Hep1, HuH7 and Hep3BX cells	CPAP	TNF-α can upregulate SUMO-1 mediated CPAP SUMOylation, which is very important for NF-κB co-activator activity.	[56] 2013
	In Vitro, MHCC97H, HepG2, HEP3B and SMMC-7721. In vivo, Four-week-old female BALB/c nude mice	UBC9, SUMOylated Mettl3 and Snail	SUMOylate Mettls can regulate the homeostasis of Snail mRNA through m6A methyltransferase activity-dependent manner. And then regulate the development of HCC.	[57] 2020
HCC proliferation and survival	in vitro, hepg2 and smmc7721 In clinical, HCC patient	NF-κB	SUMO2/3 and P65 are co-expressed and co-localised in HCC.	[58] 2015
	In vitro, MHCC97L &SMMC-7721 cells	Cbx4	CBx4 can control HIF-1α, thereby upregulating the angiogenesis of HCC. SUMO’s E3 ligase controls this activity.	[49] 2014
	In vitro, HepG2 cells	WWOX	WWOX can help SENP2 stabilisation of β-catenin.	[59] 2014
	In-clinic, Patients HCC cells	HIF-1/2α	SENP1 can de-SUMOylation HIF-1α and increase its stability.	[60] 2016
	In vitro, HepG2 cells	NF-κB	SUMO1 and P65 are co-expressed and co-localised. Up-regulation of SUMO1 can upregulate NF-κB activity and promote the progression of HCC.	[61] 2016
	In vivo, Male BALB/c nude mice	ATR	SUMOylation of ART can upregulate the proliferation of HCC cells.	[62] 2016
	In vitro, human LCSC In-clinic, 60 HCC patients.	TRF2	HULC can enhance phosphorylation, thereby suppressing SUMOylation.	[63] 2016
	in vitro, HEK293T, L02 normal human hepatic cells, HepG2 human hepatocellular	Lats	The SUMOylation of Lats1 can upregulate Hippo signalling.	[64] 2017
	in vitro, Huh-7, Hep G2 and PLC/PRF/5 and MLP-29	lkb1	SUMO2-mediated SUMOylation of LKB1 can affect its location in cells and promote carcinogenesis.	[65] 2020
HCC migration, invasion and metastasis	In vitro, PHHs, MHCC-97H and HCCLM3. In-clinic, human HCC patient	P53 hnRNP SUMOlyation	The SUMOylation of hnRNP k can activate the P53 signal, thereby inhibiting tumour progression.	[66] 2020
	In vitro, Huh-7 and HepG-2In-clinic, Cancer tissues and Paracancerous tissues for 38 patients.	NF-κB	The SUMOylation of NEMO can upregulate the activity of NF-κB, thereby promoting the invasion of HCC.	[24] 2020
	In Vivo, MANF-KO [knockout] mice In-Clinic, Human HCC cells	NF-κB	SUMO1 co-localises with P65 and MANF. Thereby promoting the interaction between P65 and MANF.	[25] 2020
	in vitro, Huh7, SK-Hep1, Hep3B, Hep3BX, HepG2 and HepG2X cells	HBx	CPAP can upregulate the SUMOylation of HBx.	[67] 2019
	in vivo, Hnf1a-null mice	mrR192/194	SUMO2 is miR194′s target.	[68] 2017
	In-clinic, HCC patient	hsp27	SUMO2/3 upregulated the HSP27 protein level.	[69] 2017
	In vivo, HBx mice	IGF-II	SUMOylation of E-cadherin causes it to degrade.	[70] 2015
	In vitro, MHCC97L cells	Cbx4	Cbx4 can upregulate the SUMOylation of HIF-1α and shape the hypoxic microenvironment.	[49] 2014
	In vitro, HepG2, Hep3B and MHCC97H and standard liver cancer cell lines THLE-2 and LO2	Exportin-5 (XPO5)	SUMOylated XPO5 down-regulating the nucleo-cytoplasm transportation of pre-miR-3184	[71] 2020
HCC tumour microenvironment	In Vitro, HCC cell line Hep3B	HIF-1α and Oct4	deSUMOylation of HIF-1α and Oct4 reduced their accumulation in the nucleus, thereby inhibiting tumour angiogenesis and stemness maintenance.	[72] 2020
	In-clinic, human blood samples	PKM2	SUMOylation of PKM2 induced its plasma membrane targeting and subsequent exosomal excretion via interactions with ARRDC1.	[73] 2020
	in vitro, Hep3B cells In-clinic, 10 HCC patients	SHh	SUMOylation of GLI protein can upregulate its hypoxia-dependent activation.	[74] 2019
	In-clinic, HCC patients.	NF-κB	Sorafenib treatment treats HCC by inhibiting the SUMOylation of p65.	[75] 2018
	In vitro, HepG2 and SMMC-7721	UBC9	UBC9 E2 conjugating enzyme is necessary for the SUMOylation process.	[69] 2017
	In vitro, HepG2 in-clinic, HCC patient	not mentioned	not mentioned	[76] 2015

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
