# Peer review of "The Role of Protein SUMOylation in Human Hepatocellular Carcinoma: A Potential Target of New Drug Discovery and Development"

_cancers, 2021, doi:10.3390/cancers13225700_

Round 1

Reviewer 1 Report

In this manuscript by Yuan et al., the authors discuss SUMOylation in human hepatocellular carcinoma.

Major comments:

  1. A paragraph on the types of post-translational modifications should be included prior to paragraph 2 in the introduction.
  2. The authors should include a Figure showing SUMO proteins and SUMOylation so it would be easier to understand the written content.
  3. Figures 1 and 2 are shown but are not referred to in the text. Please correct this.
  4. Quality of Figure 1 is not good enough for publication. The axes are barely visible. Please replace with a better quality image.
  5. The section on SUMO drugs in clinical trials has to be moved into section 4 and not be in the Discussion.

Minor comments:

  1. HCC abbreviation is included in the Abstract and Introduction without initially spelling out what it is.
  2. Paragraphs in Lines 265-267 have to be reworded.

Author Response

Dear Editor:                                                         Nov. 09, 2021

Manuscript ID: cancers-1439124

The role of Protein SUMOylation in human hepatocellular carcinoma: a potential target of new drug discovery and development

Thank you for your and reviewers’ comments concerning our above-mentioned manuscript submitted to Cancers. Those comments are valuable and helpful for improving our paper. We have studied comments carefully and made substantial revisions. The amendments are highlighted in red throughout the paper. Here are the details:

Responds to the reviewer’s comments:

  • Reviewer #1:

Response to comment:

  1. Major comments:
  2. A paragraph on the types of post-translational modifications should be included prior to paragraph 2 in the introduction.
  3. The authors should include a Figure showing SUMO proteins and SUMOylation so it would be easier to understand the written content.
  4. Figures 1 and 2 are shown but are not referred to in the text. Please correct this.
  5. Quality of Figure 1 is not good enough for publication. The axes are barely visible. Please replace with a better quality image.
  6. The section on SUMO drugs in clinical trials has to be moved into section 4 and not be in the Discussion.

Response: Thank you very much for your comments. I have made the following changes to the manuscript: a) A brief paragraph of post-translational modification has been added to the introduction; b) a block diagram of SUMOylation has been added to the graphical abstract; c) Figure 1 and 2 have been cited in the main text according to the "GEPIA" specification; d) We have replaced Figure 1 with a higher resolution image; e) and moved the discussion of clinical trials to the appropriate place.

  1. Minor comments:
  2. HCC abbreviation is included in the Abstract and Introduction without initially spelling out what it is.
  3. Paragraphs in Lines 265-267 have to be reworded.

Response: Thank you for your comments. We apologize for these mistakes. a) I have added the full name of "HCC" in the Abstract and Introduction accordingly; b) I also have reorganized the language of lines 265-267 to make the sentences more fluent and make sense. In addition, the sentences and grammar of the full text have been revised to make the article more logical.

We have carefully read the recommendations of the Reviewers/Editors and revised the manuscript. Thank you very much for your enthusiasm, meticulous and fast work. I hope that the content of the revised article will be improved and can meet your requirements. Once again, thank you very much for your valuable comments and suggestions.

Best regards,

Ning Wang

School of Chinese Medicine

The University of Hong Kong

[email protected]

Reviewer 2 Report

Thanks for the effort from the authors. Some critical issues should be addressed before the article could be published.

Major issues:

1.  Line 37-39

I’m not sure that sorafenib is the “first drug” for unresectable HCC. But certainly, it’s the first “tyrosine kinase inhibitor” or “small molecular inhibitor” for unresectable HCC. Would the authors like to confirm this or rephrase it?

Besides, in the past years, a growing number of therapeutic strategies are listed in the treatment guideline, such as immunotherapy with antiangiogenic therapy (atezolizumab and bevacizumab). Authors would better to mention about these agents, because they are also “effective options”.

2.  Line 59

The title of section 2 is “SUMO protein expression in HCC”. However, in this full section, I did not find any description about “SUMO expression in HCC”. The entire section is all about SUMOylation and its associated enzymes. I would suggest the authors to change this title.

3.  Line 138-146

In the paragraph, the authors cited the study about SUMO1P3. However, SUMO1P3 is a long non-coding RNA (lncRNA), and it will not translate to SUMO protein at all. I think SUMO1P3 is not fit the scope of your study. I would suggest to delete the paragraph about SUMO1P3. Besides, the summarized description of SUMO1P3 in Table 1 may also be removed.

4.   Line 242

The authors mentioned that patients with higher levels of MANF have “worse” DFS and OS. However, the description is totally against the original study reported by Liu et al. I would suggest the authors to check the original reference again.

5.   Figure 1 and Figure 2

I would suggest the authors to add some description about the mRNA expression level of SUMOs and their survival significance into the section 3. “The expression and clinical significance of SUMOs proteins in HCC”.

The content of Figure 1 and 2 is more appropriate to be placed in this section. Besides, is the “protein” level of SUMOs also over-expressed in HCC tumor part?

Minor issues:

1. many typos

Line 43   cascad -> cascade

Line 70   SUMOlated -> SUMOylated

Line 231  4.3. UMOylation -> SUMOylation

Line 236  “mesencephalic” astrocyte derived neurotrophic factor (MANF)

Line 295  hepatocellular carcinoma -> HCC
Line 345  plays an -> a

Table 1  about the effects of reference 66: 
      SUMOylazed Mettls -> SUMOylated Mettl3

Table 1 about the reference 72:
      mrR192 -> miR192

Figure 1  figure legend
line 2: figure below ->figure above

line 4:  GEPOA -> GEPIA

2. Would the authors like to unify the writing style of English words, such as “P65 or p65”, “tumour or tumor”?

3.  Some Ambiguous words:

Line 114-115

SUMO2/3 proteins mainly separate SENP3 and SENP5.
- > SUMO2/3 proteins are mainly separated by SENP3 and SENP5.

4.  Figure legend of Figure 1:

line 4: suggest use “ HCC tumor tissue” instead of “liver cancer cells” and
        “ normal liver tissue” instead of “normal cells”
   Because the result is from experiment of bulk sequencing but not single cell sequencing. So we can not know the exact SUMO expression level in liver cancer cells.  

Author Response

Dear Editor:                                                         Nov. 09, 2021

Manuscript ID: cancers-1439124

The role of Protein SUMOylation in human hepatocellular carcinoma: a potential target of new drug discovery and development

Thank you for your and reviewers’ comments concerning our above-mentioned manuscript submitted to Cancers. Those comments are valuable and helpful for improving our paper. We have studied comments carefully and made substantial revisions. The amendments are highlighted in red throughout the paper. Here are the details:

Responds to the reviewer’s comments:

  • Reviewer #2:

Thank you very much for your comments. I have made some corrections to the manuscript based on your comments. The following is a summary of the changes.

  1. Line 37-39

I’m not sure that sorafenib is the “first drug” for unresectable HCC. But certainly, it’s the first “tyrosine kinase inhibitor” or “small molecular inhibitor” for unresectable HCC. Would the authors like to confirm this or rephrase it?

Besides, in the past years, a growing number of therapeutic strategies are listed in the treatment guideline, such as immunotherapy with antiangiogenic therapy (atezolizumab and bevacizumab). Authors would better to mention about these agents, because they are also “effective options”.

Response: Thank you for your comments. a) Sorafenib is the first TKI approved by the FDA for the first-line treatment of HCC[1] b) I have reviewed some clinical papers published from 2020 to 2021 and found that the timeliness of previous references were not ideal. I have modified this statement and briefly introduced these TKIs (such as Lenvatinib and Sorafenib) and immunotherapy with antiangiogenic therapy (atezolizumab and bevacizumab).

  1. Line 59

The title of section 2 is “SUMO protein expression in HCC”. However, in this full section, I did not find any description about “SUMO expression in HCC”. The entire section is all about SUMOylation and its associated enzymes. I would suggest the authors to change this title.

Response: Thank you for your comments. I have changed the title of the second paragraph to "The Dynamics of SUMOylation in HCC" which is more linked to the content.

  1. Line 138-146

In the paragraph, the authors cited the study about SUMO1P3. However, SUMO1P3 is a long non-coding RNA (lncRNA), and it will not translate to SUMO protein at all. I think SUMO1P3 is not fit the scope of your study. I would suggest to delete the paragraph about SUMO1P3. Besides, the summarized description of SUMO1P3 in Table 1 may also be removed.

Response: Thank you for your comments. I have replaced the example of SUMO1P3 and deleted the relevant content in the table.

  1. Line 242

The authors mentioned that patients with higher levels of MANF have “worse” DFS and OS. However, the description is totally against the original study reported by Liu et al. I would suggest the authors to check the original reference again.

Response: Thank you for your comments. I am sorry for the mistake.  It should be patients with higher levels of MANF have “better” DFS and OS. I have made the revision accordingly.

  1. Figure 1 and Figure 2

I would suggest the authors to add some description about the mRNA expression level of SUMOs and their survival significance into the section 3. “The expression and clinical significance of SUMOs proteins in HCC”.

The content of Figure 1 and 2 is more appropriate to be placed in this section. Besides, is the “protein” level of SUMOs also over-expressed in HCC tumor part?

Response: Thank you for your comments. According to the existing reports, both SUMOs protein and mRNA are overexpressed in HCC[2-5].

As mentioned in the reply in "Comment 3" above, I have removed the part of SUMO1P3 and replaced it with the expression level of SUMOs protein and mRNA and the description of its survival significance.

Regarding the position of the figures, I agreed to move Figures 1 and 2 to the third part.

  1. Minor issues:
  2. many typos
  3. Would the authors like to unify the writing style of English words, such as “P65 or p65”, “tumour or tumor”?
  4. Some Ambiguous words:

Line 114-115

SUMO2/3 proteins mainly separate SENP3 and SENP5.
- > SUMO2/3 proteins are mainly separated by SENP3 and SENP5.

  1. Figure legend of Figure 1:

line 4: suggest use “ HCC tumor tissue” instead of “liver cancer cells” and
        “ normal liver tissue” instead of “normal cells”
   Because the result is from experiment of bulk sequencing but not single cell sequencing. So we can not know the exact SUMO expression level in liver cancer cells.  

Response: Thank you very much for your suggestions. I have checked the whole manuscript and corrected them all as you suggested.

Reference:

  1. Rinaldi, L.; Vetrano, E.; Rinaldi, B.; Galiero, R.; Caturano, A.; Salvatore, T.; Sasso, F. C., HCC and Molecular Targeting Therapies: Back to the Future. Biomedicines 2021, 9, (10).
  2. Lee, J. S.; Thorgeirsson, S. S., Genome-scale profiling of gene expression in hepatocellular carcinoma: classification, survival prediction, and identification of therapeutic targets. Gastroenterology 2004, 127, (5 Suppl 1), S51-5.
  3. Guo, W. H.; Yuan, L. H.; Xiao, Z. H.; Liu, D.; Zhang, J. X., Overexpression of SUMO-1 in hepatocellular carcinoma: a latent target for diagnosis and therapy of hepatoma. J Cancer Res Clin Oncol 2011, 137, (3), 533-41.
  4. Tao, Y.; Li, R.; Shen, C.; Li, J.; Zhang, Q.; Ma, Z.; Wang, F.; Wang, Z., SENP1 is a crucial promotor for hepatocellular carcinoma through deSUMOylation of UBE2T. Aging (Albany NY) 2020, 12, (2), 1563-1576.
  5. Chen, J.; Chen, C.; Lin, Y.; Su, Y.; Yu, X.; Jiang, Y.; Chen, Z.; Ke, S.; Lin, S.; Chen, L.; Zhang, Z.; Zhang, T., Downregulation of SUMO2 inhibits hepatocellular carcinoma cell proliferation, migration and invasion. FEBS Open Bio 2021, 11, (6), 1771-1784.

We have carefully read the recommendations of the Reviewers/Editors and revised the manuscript. Thank you very much for your enthusiasm, meticulous and fast work. I hope that the content of the revised article will be improved and can meet your requirements. Once again, thank you very much for your valuable comments and suggestions.

Best regards,

Ning Wang

School of Chinese Medicine

The University of Hong Kong

[email protected]

Round 2

Reviewer 1 Report

Major comments:

  1. Figures 1 and 2 should be moved to the main body of the manuscript and should be referred to in the text of the manuscript.
  2. The writing in Figure 2 is too difficult to read.
  3. Line 71 is incomplete.
  4. The legend in Figure 3 is incomplete and the axes and p value are not stated. 
  5. The legend in Figure 4 has to be expanded.
  6. The referencing style in the manuscript has to be kept constant and not change in the manuscript.

Author Response

Dear Editor:                                                         Nov. 12, 2021

Manuscript ID: cancers-1439124

The role of Protein SUMOylation in human hepatocellular carcinoma: a potential target of new drug discovery and development

Thank you for your and reviewers’ comments concerning our above-mentioned manuscript submitted to Cancers. Those comments are valuable and helpful for improving our paper. We have studied comments carefully and made substantial revisions. The amendments are highlighted in red throughout the paper. Here are the details:

Responds to the reviewer’s comments:

  • Reviewer #1:

Response to comment:

Major comments:

  1. Figures 1 and 2 should be moved to the main body of the manuscript and should be referred to in the text of the manuscript.

Figure 1 and Figure 2 have been moved to the corresponding parts of the main text, and citations have been added.

  1. The writing in Figure 2 is too difficult to read.

The text in Figure 2 (now Figure 1) has been rewritten, and long sentences have been reduced to improve readability.

  1. Line 71 is incomplete.

The sentence of line 71 was rewritten to make it more complete.

  1. The legend in Figure 3 is incomplete and the axes and p value are not stated. 

Added P value and ordinate description in Figure 3.

  1. The legend in Figure 4 has to be expanded.

The description in Figure 4 has been expanded, and the cut-off value and brief conclusion have been added.

  1. The referencing style in the manuscript has to be kept constant and not change in the manuscript.

Use ENDNOTE to unify the citation format. Manually corrected the fonts cited in the text.

We have carefully read the recommendations of the Reviewers/Editors and revised the manuscript. Thank you very much for your enthusiasm, meticulous and fast work. I hope that the content of the revised article will be improved and can meet your requirements. Once again, thank you very much for your valuable comments and suggestions.

Best regards,

Ning Wang

School of Chinese Medicine

The University of Hong Kong

[email protected]

Reviewer 2 Report

Line 187, 192, 199, 202

Figure 3 title and Figure 4 title

In above section, the authors use GEPIA to conclude that SUMOs have higher expression in liver cancer and confer poor prognosis. However, GEPIA is a database of transcriptomics, e.g. mRNA, not of proteomics. I would suggest the authors to replace the word “protein” by “mRNA” in above description.

By the way, if the authors try to analyze protein level of SUMOs in liver cancer, the CPTAC (The National Cancer Institute’s Clinical Proteomic Tumor Analysis Consortium) is a source of choice for proteomic data.

Line 288

4.3. UMOylation -> SUMOylation

Author Response

Dear Editor:                                                         Nov. 12, 2021

Manuscript ID: cancers-1439124

The role of Protein SUMOylation in human hepatocellular carcinoma: a potential target of new drug discovery and development

Thank you for your and reviewers’ comments concerning our above-mentioned manuscript submitted to Cancers. Those comments are valuable and helpful for improving our paper. We have studied comments carefully and made substantial revisions. The amendments are highlighted in red throughout the paper. Here are the details:

Responds to the reviewer’s comments:

  • Reviewer #2:

Thank you very much for your comments. I have made some corrections to the manuscript based on your comments. The following is a summary of the changes.

  1. Line 187, 192, 199, 202 Figure 3 title and Figure 4 title

 In above section, the authors use GEPIA to conclude that SUMOs have higher expression in liver cancer and confer poor prognosis. However, GEPIA is a database of transcriptomics, e.g. mRNA, not of proteomics. I would suggest the authors to replace the word “protein” by “mRNA” in above description.

By the way, if the authors try to analyze protein level of SUMOs in liver cancer, the CPTAC (The National Cancer Institute’s Clinical Proteomic Tumor Analysis Consortium) is a source of choice for proteomic data.

 Thank you very much for your correction. I have replaced all the "protein" in this part with "mRNA". I also searched on CPTAC to try to expand the graph of the protein part. Unfortunately, there is currently no research related to SUMO and HCC on the website. Still,  thank you for your suggestions and guidance.

  1. Line 288

4.3. UMOylation -> SUMOylation

Corrections have been made and will be paid more attention in the future.

We have carefully read the recommendations of the Reviewers/Editors and revised the manuscript. Thank you very much for your enthusiasm, meticulous and fast work. I hope that the content of the revised article will be improved and can meet your requirements. Once again, thank you very much for your valuable comments and suggestions.

Best regards,

Ning Wang

School of Chinese Medicine

The University of Hong Kong

[email protected]
